# Insights on Current Strategies to Decolonize the Gut from Multidrug-Resistant Bacteria: Pros and Cons

**DOI:** 10.3390/antibiotics12061074

**Published:** 2023-06-19

**Authors:** Natalia Roson-Calero, Clara Ballesté-Delpierre, Javier Fernández, Jordi Vila

**Affiliations:** 1Barcelona Institute for Global Health (ISGlobal), 08036 Barcelona, Spain; natalia.roson@isglobal.org (N.R.-C.); clara.balleste@isglobal.org (C.B.-D.); 2Department of Basic Clinical Practice, School of Medicine, University of Barcelona, 08036 Barcelona, Spain; 3CIBER de Enfermedades Infecciosas (CIBERINFEC), Instituto Salud Carlos III, 28029 Madrid, Spain; 4Liver ICU, Liver Unit, Hospital Clinic, University of Barcelona, IDIBAPS and CIBERehd, 08036 Barcelona, Spain; jfdez@clinic.cat; 5European Foundation for the Study of Chronic Liver Failure (EF-Clif), 08021 Barcelona, Spain; 6Department of Clinical Microbiology, Biomedical Diagnostic Center, Hospital Clinic, 08036 Barcelona, Spain

**Keywords:** decolonization, multidrug-resistant bacteria, gut microbiome, bacteriotherapy, phage therapy, CRISPR-Cas, fecal microbiota transplantation

## Abstract

In the last decades, we have witnessed a steady increase in infections caused by multidrug-resistant (MDR) bacteria. These infections are associated with higher morbidity and mortality. Several interventions should be taken to reduce the emergence and spread of MDR bacteria. The eradication of resistant pathogens colonizing specific human body sites that would likely cause further infection in other sites is one of the most conventional strategies. The objective of this narrative mini-review is to compile and discuss different strategies for the eradication of MDR bacteria from gut microbiota. Here, we analyse the prevalence of MDR bacteria in the community and the hospital and the clinical impact of gut microbiota colonisation with MDR bacteria. Then, several strategies to eliminate MDR bacteria from gut microbiota are described and include: (i) selective decontamination of the digestive tract (SDD) using a cocktail of antibiotics; (ii) the use of pre and probiotics; (iii) fecal microbiota transplantation; (iv) the use of specific phages; (v) engineered CRISPR-Cas Systems. This review intends to provide a state-of-the-art of the most relevant strategies to eradicate MDR bacteria from gut microbiota currently being investigated.

## 1. Introduction

In the last decades, we have witnessed a steady increase in infections caused by multidrug-resistant (MDR) bacteria [1]. MDR can be described as the lack of susceptibility of an isolate to at least one agent in three or more chemical antimicrobial classes [2]. Action plans to combat antimicrobial resistance have been launched in most countries. Most include a list of potential interventions to decrease the emergence of MDR bacteria and their spread [3,4]. Different actions should be considered in the hospital setting to control the spread of MDR bacteria. It is essential to highlight the decolonization or sensitization of these MDR bacteria [4]. In addition, the discovery and development pipeline of new strategies to fight against MDR bacteria, either the development of traditional (new antibiotics) or non-traditional (i.e., endolysins or phages, monoclonal antibodies against virulence factors, etc.), is scarce [5].

Moreover, with the increase of MDR bacteria, the options to provide an adequate empirical antimicrobial agent will decrease. In this sense, a rapid diagnosis of the etiological agent of the infection is needed to guide antibiotic strategies adequately. Another key aspect to consider is the prevention of infections caused by MDR bacteria is still required. One strategy is the eradication of the resistant pathogen colonizing specific human body sites that would likely cause further infection in other sites. A meta-analysis evaluating patient washing with chlorhexidine washcloths and wipes in a hospital setting showed a reduction in the hospital-acquired infection rate, an effect which was more evident in Gram-positive infections [6]. In addition to the skin, the gut is also an essential reservoir of MDR bacteria. This is relevant in hospitalized patients, especially in cases where the patient has gut dysbiosis with the predominance of a specific bacterial species, which can be associated with subsequent bacteraemia [7]. The present review is aimed to analyse the presence of MDR bacteria within the gut microbiota, their clinical impact, and the level of development of different strategies to eradicate them, from most to least traditional.

## 2. Prevalence of Multidrug-Resistant Bacteria in Gut

### 2.1. In the Community 

Although local epidemiology studies focusing on MDR carriage among healthy populations have been conducted in different settings, population types and screening methods are heterogeneous, so poor global estimates have been reported. A recent systematic review and meta-analysis evaluated the prevalence of fecal carriage of extended-spectrum β-lactamase-producing (ESBL) *Escherichia coli* among healthy individuals. They found an alarming 10-fold increase in the global community prevalence over the last 20 years: from 2.6% in 2001–2005 to 26.4% in 2016–2020 [8]. 

At the country level, however, huge differences have been reported so far. In Europe, a large study was conducted in The Netherlands (2014–2017) in which the epidemiology of carriage of ESBL-producing (ESBL-E) and carbapenemase-producing *Enterobacterales* (CPE) was analysed. The study included fecal samples from 4177 people, and results showed 5% of the samples were positive for ESBL-E, whereas no CPE was identified [9]. In Germany, a study was conducted among the paediatric population upon admission at Altona Children’s Hospital (AKK) in Hamburg in 2018–2019. Sampling on 3964 patients resulted in a 3.64% MDR Gram-negative carriage in this population [10].

More alarming reports were seen in two studies conducted in France: the first one was conducted in 2006 and included 332 people. Only 0.6% were positive for ESBL-producing *E. coli*. Five years later, a ten-fold increase in MDR carriage was observed in a similar population (345 people were examined) [11]. Higher data rates were seen in a study conducted among 102 healthy students at California University identified 72% of Gram-negative bacteria carrying β-lactamase genes, dihydrofolate reductase genes and aminoglycoside resistance genes [12]. In this line, a recent report from Taiwan, including 187 healthy participants, showed 41.4% (77/186) carrying third-generation cephalosporin-resistant *E. coli* or *Klebsiella pneumoniae* [13]. 

Although clear limitations exist on data available and more data is needed to estimate the global burden of MDR carriage, reports published so far indicate an alarming increase in the MDR rates in the community.

### 2.2. In the Hospitals

Several factors influence the gastrointestinal colonization of MDR bacteria in hospital settings that may lead to nosocomial infections. It has been reported that long stays at hospitals or taking antibiotics three months before surgery negatively influence the fecal colonization of ESBL-E [14]. However, Wieërs et al., 2021 suggest that patients treated with probiotics at the moment of their entrance to the hospital help to reduce the colonization of the digestive tract with MDR pathogens like *Pseudomonas aeruginosa* [15], which has also been confirmed by other studies [16,17]. A meta-analysis performed by Vink et al., 2020, revealed that antibiotic usage, especially belonging to carbapenems, played a key role in the acquisition of MDR Gram-negative bacteria during a patient’s hospital admission, being *K. pneumoniae* the one presenting a higher risk when it comes to patient-to-patient transmission in European and North American hospitals [18]. In addition, the presence of MDR bacteria on the surface of surgery instruments, hygiene-related items, or equipment was studied in a hospital in Brazil. Among all the isolates obtained, more than 93% were antimicrobial resistant, with high resistance rates for ampicillin and trimetoprim-sulphametoxazol in both Gram-negative and Gram-positive bacteria, and for amoxicillin in the case of Gram-negative bacteria, what manifests the important role that hospital environment may play regarding gut colonization by nosocomial MDR bacteria [19]. Moreover, studying the burden of AMR bacteria in hospitals from another perspective, it has been demonstrated that wastewater coming from hospitals is enriched with microbiota associated with the human gut compared to the rest of urban wastewater [20]. Hospital wastewaters contain significantly higher loads of antimicrobial resistance genes, including carbapenem, tetracycline or sulphonamide resistance genes and genetic mobile elements, which entails an essential source of dissemination of these genes to the environment, too [20,21].

### 2.3. Clinical Importance of Carrying MDR Bacteria in the Gut for the Development of Infections

Several papers have shown the importance of MDR-resistant bacteria in gut microbiota as the origin of extraintestinal infections, such as recurrent urinary tract infections or as a cause of hospital-acquired bloodstream infections and pneumonia [22]. In the third case, it was believed that most of the gut bacteria in the lungs of healthy people incorporated them by micro aspiration. However, a study by Dickson et al. [23] showed that gut-associated bacteria, undetectable via conventional culture techniques, caused the enrichment of the lung microbiome in experimental sepsis and human acute respiratory disease syndrome. They reported that the lower gastrointestinal tract, rather than the upper respiratory tract, was the likely source of bacteria in the lung, suggesting that alternative pathways for bacteria to reach the lung from the gut might exist. Intestinal colonization with ESBL-E is a high-risk factor for the development of ESBL-E infections. In addition, the environmental contamination and transmission of ESBL-E to other patients in the hospital setting have also been demonstrated [24,25,26]. It has been suggested that the presence of ESBL-E in the throat and/or high densities of colonization in the rectum might be a tool to estimate the risk of subsequent ventilated-associated pneumonia caused by ESBL-E, thus, to initiate an empiric antibiotic therapy in intensive care units (ICUs) patients which would more likely be appropriate [27]. However, another study [28] showed that Gram-negative rectal colonization tends to be more strongly associated with subsequent ICU-acquired Gram-negative infections than Gram-negative respiratory tract colonization.

Similarly, in a systematic review and meta-analysis, Alevizakos and colleagues showed that cancer patients with ESBL-E colonization were 12.98 times (95% CI 3.91–43.06) more likely to develop a bloodstream infection with ESBL-E during their hospitalization compared with non-colonized patients [29].

The infections caused by carbapenem-resistant *Enterobacterales* and carbapenem-resistant non-fermentative Gram-negative bacilli such as *P. aeruginosa* and *A. baumannii* are steadily increasing [30]. The abovementioned microorganisms have been identified as common rectal and respiratory tract colonizers [31,32], and it is associated with the development of infection [33]. In a systematic review of risk factors associated with infection progression in adult patients with respiratory tract or rectal colonization by carbapenem-resistant Gram-negative bacteria, the authors found that among the risk factors analyzed, previous exposure to antibiotic therapy or previous carbapenem use was relevant for patients with carbapenem-resistant Gram-negative bacilli respiratory tract and rectal colonization. They concluded that the early identification of colonized patients with carbapenem-resistant bacilli, with a high risk of infection development, can favour effective therapy and improve health outcomes [34].

The potential role of *P. aeruginosa* intestinal colonization in the subsequent development of infections in patients in an ICU has also been studied; one-hundred seventy-nine (43%) out of the 414 studied patients were colonized with *P. aeruginosa.* The probability of *P. aeruginosa* infection 14 days after ICU admission was 26% for patients colonized *versus* 5% for noncolonized (*p* < 0.001). Therefore, the knowledge of *P. aeruginosa* colonization may help to initiate appropriate empirical therapy [35]. In two additional studies, analysing the impact of being colonized by *P. aeruginosa* in patients admitted to an ICU to develop an infection by the colonized strain further, the data was: 28.2% for patients colonized with *P. aeruginosa* versus 4.2% in non-colonized [36] and 62% in *P. aeruginosa* colonized patients versus 4% in non-colonized [37].

A recent study by Sun Y et al. (2021) analyzed the association between the relative abundance of *Klebsiella pneumoniae* and subsequent infections, founding that the relative abundance of this bacteria determined by qPCR was higher in cases (15.7%) than in controls (1.01%) and suggest that a relative abundance higher than 22% was highly associated with infections. Therefore, colonization density could be a better marker for potential further infection development than the mere presence of *K. pneumoniae* [38].

## 3. Strategies to Decolonize the Gut from MDR Bacteria 

AMR reduction strategies are limited to control measures and infection prevention based on active surveillance for early identification of the carriers, which few times include microbiome profiling through metataxonomic analyses [39]. Based on the abovementioned studies showing that gut colonization with multidrug-resistant microorganisms is an essential risk factor for further infections caused by colonizing pathogens, their specific gut elimination could be an attractive preventive approach. As described by ESCMID-EUCIC, a decolonizing therapy is defined as any measure leading to a reduction of detectable MDR bacteria carried at any site [40]. As a relatively new concept, strategies based on gut decolonization still have low levels of evidence, and robust recommendations cannot be provided yet [41] if not part of a registered clinical trial. Fortunately, decolonization therapies are gaining support in the field, and several strategies are currently under clinical study to carry out this elimination (Table 1). Here, we detail the most important ones.

### 3.1. Antibiotics

The application of non-absorbable antibiotics to the mouth and stomach, together with a course of broad-spectrum intravenous antibiotic(s), has been used as selective decontamination of the digestive tract (SDD). This was designed a few decades ago to decrease the morbidity and mortality from infections, mainly in ICUs. However, there is no consensus regimen for SDD; different antibiotics may be applied in different ways, and some regimens include parenteral antibiotic application, and others do not. SDD regimens appear effective for preventing ICU-acquired infections [42,43,44]. A meta-analysis that included 64 studies (of which 47 were randomized controlled trials) analysed the short-term impact of the SDD on the development of antimicrobial resistance in 35 of them. When comparing patients in the intervention groups (those receiving SDD) with those in the control groups, there were no statistically significant differences in the prevalence of methicillin-resistant *Staphylococcus aureus* colonization or infection, vancomycin-resistant Enterococci and aminoglycosides or fluoroquinolone-resistance Gram-negative bacilli [45]. It is important to remark that this meta-analysis was performed with studies published before 2013. The situation regarding the prevalence of ESBL and/or carbapenemase-producing Gram-negative bacilli has changed. 

However, the key point is whether the SDD accelerate the rise of resistance in gut microbiota. In this sense, the emergence of a colistin-resistant *K. pneumoniae* within an ICU was attributed to the introduction of oral colistin as part of an SDD regimen in a study published in South Africa [46]. In a retrospective analysis of a German outbreak, SDD with colistin and gentamicin as an oral solution and gel was used in 14 patients with proven carriage of KPC-producing *K. pneumoniae*. Eliminating KPC-producing *K. pneumoniae*, defined by three PCR screens 48 h apart, was found in six treated patients (43%) and 30% of controls. Resistance to colistin and gentamicin in post-treatment isolates of *K. pneumoniae* rose 19% and 45%, respectively, compared with controls [47]. 

In addition, another important point is the impact of SDD on the gut microbiota. After SDD, there is a significant decrease in different phylum (*Proteobacteria, Actinobacteria, Firmicutes* and *Fusobacteria*) [48]. A study showed that SDD notably impacts the composition of the anaerobic intestinal microbiota; the number of *Fecalibacterium prausnitzii*-group of bacteria is significantly reduced during the SDD regimen compared to the standard of care regimen [49]. It is known that *F. prausnitzii* plays an important role since they are butyrate-producing bacteria with anti-inflammatory properties [50].

As an overall statement, the use of SDD in ICUs with low levels of antimicrobial resistance is justified, but not in ICUs with a high prevalence of MDR bacteria. ESCMID-EUCIC clinical guidelines on decolonizing MDR Gram-negative bacteria carriers do not recommend routine decolonization [40]. Moreover, its impact on the gut microbiota should be taken into consideration. A potential alternative would be to perform fecal microbiota transplantation after the SDD.

### 3.2. Bacteriotherapy: Probiotics, Prebiotics and Synbiotics

According to the International Scientific Association of Probiotics and Prebiotics (ISAPP), a probiotic can be defined as “live microorganisms which, when administered in adequate amounts, confer a health benefit on the host” [51]. On the contrary, prebiotics refers to nondigestible compounds specifically fermented by commensal microbiota that help increase the diversity within gut commensals by promoting a favourable growth environment [52]. Both combined result in the so-called symbiotic treatments [53]. Since their appearance as nutritional complements for gut microbiota modulation, few studies have been published that relate their intake with a displacement of the host’s resistome [54]. Current human Randomized Controlled Trials (RCTs) revealed no statistical significance in the eradication of intestinal carriage with MDR organisms compared to the placebo group, as reported by Karbalaei et al., 2022 [55]. Even ESCMID/EUCIC guidelines from 2019 recognize insufficient evidence to support the use of probiotics to this end, so the true role of probiotics in eradicating MDR bacteria remains uncertain. Essential aspects like dosages, frequency or administration route have not been sufficiently assessed, which may be why their clinical relevance remains unclear. 

However, some studies managed to succeed in the use of this therapy. In this sense, one of the most promising results so far reported by different studies probes the protective effect of the probiotic strain *Lactobacillus rhamnosus* GG over vancomycin-resistant *Enterococci* (VRE), which was eliminated in more than 90% of participants, with clear differences with respect to the placebo groups, despite previous antibiotic treatments that may affect the microbiota composition [56,57]. However, as revealed by Boysen, L. and colleagues, [58] no significant effect was found in using this same species to prevent and reduce the risk of ESBL-E acquisition and posterior colonization on travelers from India. Neither did *Lactobacillus plantarum* improve the outcome of patients admitted to ICU, in terms of days alive and out of the hospital to Day 60, in a recent randomized and placebo-controlled clinical trial [59]. Another RCT tested the probiotic Vivomixx^®,^ where patients from each arm were administered two sachets (9.0 × 1011 live bacteria) twice daily for two months. In the end, only 12.5% of the patients taking probiotics met the primary goal, which was to eradicate the gut from ESBL-producing *Enterobacterales* at the end of 1-year follow-up, concluding that there was no evidence of probiotics being superior to placebo in decolonizing chronic carriers [60].

In contrast with these results, in 2022, a study performed by Guitor A.K. and colleagues showed the potential of *Bifidobacterium* spp. and *Lactobacillus* spp. to displace antimicrobial resistance genes (ARGs) from *Enterobacterales* in the preterm infant gut. In this report, children not receiving probiotics were more likely to host AMR, mainly associated with previously taken antibiotics and their mechanism of inactivation [61]. This contrast in the results highlights the lack of consensus regarding using probiotics to decolonize the gut from MDR bacteria. In addition, bacteraemia derived from *Lactobacillus* administration has been reported by different studies [62,63], which increases the side effects of probiotic use and subsequently reduces the safety of these supplements, especially when it comes to patients with pathologies involving enteric mucosal barrier injury or immunosuppression.

Given the uncertainty, new-generation probiotics aim to widen the variety of strains available for probiotic supply. Although available data is still limited, some of them have been studied in preclinical trials and include species like *Faecalibacterium prausnitzii*, *Bacteroides uniformis*, *Akkermansia muciniphila* or the Clostridia clusters IV, XIVa, and XVIII [53]. Nonetheless, in 2023, nature-inspired artificial probiotics based on artificial enzyme-dispersed covalent organic frameworks (COFs) were developed by Deng et al. [64] as a new strategy to modulate the gut microbiota. While originally, COFs-based probiotics have been thought to help suppress intestinal inflammation and regulate the immune system. The authors also propose its use in the fight against MDR infections.

The other main approach for bacteriotherapy is prebiotics, which represents an advantage concerning probiotics as they do not contain living organisms. This group includes inulin, fructooligosaccharides, galactooligosaccharides or lactulose, which are not absorbed until arriving in the colon and can be fermented into short-chain fatty acids, among others, especially by lactobacilli and bifidobacterial, improving the environment for the growth of commensal bacteria. Unfortunately, no studies have been published yet regarding the use of prebiotics for MDR bacteria decolonization. Synbiotics, on the contrary, have been reported to help decrease the prevalence of colonization by *Clostridioides difficile* in healthy infants, thanks to a combination of *Bifidobacterium*, *B. breve* M-16V and the prebiotics scGOS/lcFOS (9:1) [65]. Therefore, synbiotics are gaining interest for the control of nosocomial *Acinetobacter baumannii*, as it has been demonstrated the protective effect of *Bifidobacterium breve* strain Yakult (BbY) and prebiotic galactooligosaccharides (GOS) against MDR strains in murine infection models [66]. Another successful example involving the use of synbiotics was reported in 2003, where the combination of vancomycin and a symbiotic composed of *B. breve*, *L. casi* and galactooligosaccharides helped to solve fulminant methicillin-resistant *S. aureus* (MRSA) enterocolitis in a 3-month-old infant [67]. 

### 3.3. Fecal Microbiota Transplantation

Fecal microbiota transplantation (FMT) consists of the administration of healthy fecal microbiota by using different approaches: instillation (in the proximal small bowel or the colon), oral capsules with lyophilized microbiota or frozen product given by colonoscopy or enema to restore a balanced microbiota into the receptor [39]. Due to the clinical evidence of its high therapeutic effectiveness (between 90% and 100%) and high safety profile, it is now a well-recognized treatment by international clinical guidelines [68,69] for recurrent *C. difficile* infection where the fundamental underlying problem is a severe dysbiosis. This occurs when there is a lack of microbiota recovery after an episode of diarrhea, most frequently in ≥65 years patients who recently received antibiotics. 

Since 2014, several studies have been conducted to evaluate the efficacy of FMT for MDR intestinal decolonization with interesting outcomes. As reported by Gargiullo et al., from 23 total reports analysed in which different strategies were conducted from FMT administration (31.6% of the treated patients received FMT via naso-duodenal route, followed by nasogastric tube in 25.3% of the cases and 16.8% received oral capsules), in 77.5% of patients a microbiological clearance of MDR bacteria on fecal samples or rectal swabs was achieved [39]. Regarding MDR bacteria eradication, data from the last follow-up period of all 23 cases showed that 60.0% showed Gram-positive bacteria eradication and 90.0% of Gram-negative bacteria [70]. The most found MDR intestinal bacteria before FMT was applied were carbapenem-resistant *Enterobacterales* and VRE, MDR *P. aeruginosa,* MRSA, *Acinetobacter and* ESBL-E [68,69,70].

The effectiveness and risk of FMT to decolonizing gut were assessed and published in a systematic review applying the Cochrane methodology that selected 5 out of 343 studies related to FMT. The study concluded that there was low-quality evidence of the ability of FMT to displace antibiotic-resistant bacteria in the gut, and the relatively low risk of the therapy was confirmed. Despite this conclusion, the evidence does indicate a potential benefit of FMT which can only be confirmed with RCTs that evaluate the superiority of this intervention over no-treatment [71]. Despite the increasing number of studies and case reports providing evidence of the efficacy of FMT for MDR decolonization, more homogeneous data from the currently ongoing RCTs and new well-designed ones are needed to standardize the methodology and establish well-defined guidelines as novel antimicrobial stewardship interventions [72]. 

### 3.4. Phages

The main phage characteristic is the specificity of the host bacteria, even to a strain level. Therefore, we can selectively direct phages to the MDR bacteria we want to remove from the gut without disrupting the remaining gut microbiota. There are several studies performed in animal models and some specific cases in humans in which it has been shown that phages can be used to treat MDR bacterial infections, including MDR *P. aeruginosa*, *A. baumannii* and VRE [73,74,75]. Hua, Y. and colleagues showed the in vivo efficacy of a specific phage administered intranasally to rescue mice from lethal *A. baumannii* lung infection in a mouse model without deleterious side effects [75].

However, the number of studies focused on using phages to decolonize the gut from MDR bacteria is still limited. In studies focused on animal models, specific decolonization of *E. coli* from the gut using bacteriophages has shown a reduction of this microorganism in gut mice [76,77,78]. However, reports showing the difficulty of achieving efficacy with phages to reduce *E. coli* digestive carriage in vivo have also been published [79,80,81].

The specificity of phages can also be considered their greatest disadvantage since once the multidrug-resistant bacteria to be eliminated is known, the susceptibility of it to the phages should be investigated, which is a cumbersome methodology. Another way to overcome this drawback is to use a cocktail of phages to target an extended range of the same bacterial species. Moreover, recent reports suggest that phage and additionally modified phage have failed to reciprocate against the MDR pathogens when applied in single doses [82]. The conditions for the optimal efficacy of phages in the gastrointestinal tract are still poorly understood, requiring additional knowledge. As exposed in a systematic review performed by Dabrowska et al. 2019, the complexity of effective phage therapy not only relies on the species to treat but on a complete list of factors that include: route of administration and dose-dependency, age of the patient and gut microbiota or secretion of digestive compounds that affect stomach acidity, both related to gut physiology [83]. 

### 3.5. CRISPR-Cas Systems

Microbiome-targeted therapies are emerging as new strategies to modify the resistome of the gut microbiome in a highly precise way, aiming to serve as an alternative to broad-spectrum antimicrobials [82,84,85,86,87,88,89]. Some of these are based on the use of CRISPR-Cas (Clustered Regularly Interspersed Short Palindromic Repeats—CRISPR associated protein) Systems that allow targeted genetic modifications and removal of AMR genes in selected species belonging to the gut population [84,85,86,87,88,90]. One of the biggest challenges of this approach is coming up with the most efficient delivery system. Current strategies include polymer nanocomposites, microparticles, phages or conjugative bacteria [91]. In a recent publication from 2021, Neil et al. 2021 [84] propose the COP system (a customizable engineered conjugative probiotic based on an *E. coli* Nissle 1917 strain harboring the CRISPR-Cas9 system in a conjugative plasmid called TP114) able to reduce the target population in more than 99.9% when tested in a mouse model. 

Engineered bacteriophage vectors, called CR-phages when carrying any CRISPR-System, act by detecting receptors located in the external bacterial membrane or capsule and insert CRISPR-conjugated genetic elements into the pathogen’s cytoplasm [82]. Several studies have also proven using in vitro constructed phagemids as efficient deliverers of sequence-specific antimicrobials, active against ESKAPE species like *S. aureus* or *E. coli* [85,86]. Apart from phagemids, temperate phages can also be used, with the advantage that they can be self-assembled without the intervention of helper plasmids. Although no CR-phages are available for this engineering technique for all gut pathogenic species, some have proven to be effective against clinically relevant species like *C. difficile*, *P. aeruginosa* or *K. pneumoniae* [82]. Additionally, not only killer CRISPR-Cas Systems are being studied [92]. The use of dCas9, a modified Cas9 unable to generate double-strand breaks, offers the possibility of modulating the population by repressing the expression of AMR genes and keeping the strain alive [87,88]. A step forward was given by Hsu et al., 2020, who described how λ phages expressing dCas9, and the rest of the components of the system can be oral delivered to target specific *E. coli* genes in mouse gut by its encapsulation in alginate, which protects the delivery [87]. CRISPR-Cas Systems offer a promising tool to selectively modify the gut’s bacterial microbiome while discriminating between commensal and pathogenic species, as it can be programmed to target specific species or inhibit the expression of selected genes. 

Combined therapies are also emerging to widen the available strategies involving CRISPR-Systems, often driven by the need for delivery tools to mobilize all the CRISPR elements into the targeted cell effectively. That is the case of the recently mentioned CR phages developed by Nath and colleagues in 2022 [82]. Another example is Gencay et al., 2023 [93], who selected eight broad-spectrum phages targeting clinical *E. coli* out of a 162-wild-type-phage library and engineered them with tail fibers and CRISPR–Cas machinery targeting specifically this species. With this study, they devise the drug product SNIPR001, based on a combination of these engineered phages, able to target bacteria in biofilms and selectively kill *E. coli* strains (4 log_10_ reduction) while being respectful with mice’s gut microbiota. Another approach combining CRISPR and an Antitoxin-Toxin (TA) system was developed and named by Wang et al., 2023 [94] ATTACK (AssociaTes TA and CRISPR-Cas to Kill multidrug-resistant (MDR) pathogens). In this case, the CRISPR System is programmed to kill a collection of *A. baumannii. If* somehow it is inactivated or suppressed, a CRISPR-regulated TA system based on small RNAs triggers cell death increasing the CRISPR curing effect. Having seen this, although essential improvements are happening in the field, in vivo biocontainment of the carriers of the system, narrow activity spectrum in the case of bacteriophages, or low conjugation ratios may still hinder their transition to the clinical application [82,88,89,90,91,92].

**Table 1 antibiotics-12-01074-t001:** Summary of the pros and cons of the different strategies discussed in this review to decolonize the gut from MDR bacteria.

PROS	STRATEGY	CONS
Effective to prevent ICU-acquired infections [42,43,44]	Use of antibiotics	High impact on gut microbiota [48,49] No consensus on its application [42,43,44] Accelerate the appearance of resistances [46,47] Not recommended for routine decolonization [40]
Promising results for the prevention and treatment of specific infections [56,66,67] Available as food supplements [51]	Bacteriotherapy	Lack of consensus [53] Bacteremia derived from probiotic intake have been detected [62,63]
There is clinical evidence of its therapeutic effectiveness and safeness [68,69] Numerous clinical studies are happening [39,68,69,70]	Fecal Microbiota Transplantation	Lack of homogenous data [72] Standardization of the methodology and guidelines is still needed [72]
High specificity for the host bacteria [73,74,75] Selective removal of MDR bacteria [75] Cocktails targeting different species can be administered [82]	Phages	High specificity can lead to low susceptibility [82] Poor knowledge about the efficacy of phages in the gastrointestinal tract
High gene specificity Allows selective modification of the gut microbiome Discriminates between commensal and pathogenic bacteria [82,84,85,86,87,88,89]	CRISPR-Cas Systems	Difficult biocontainment Lack of CR-Phages for all gut pathogens, the narrow activity spectrum Low in vivo conjugation ratios [82,84,85,86,87,88,89]

## 4. Conclusions

This review intends to provide state-of-the-art strategies to eradicate MDR bacteria from gut microbiota that are currently being investigated. Although no systematic methodology has been used to develop this review, the information provided delivers the current “perspective” of the potential use of the different approaches to decolonize MDR bacteria from the gut. All the above-mentioned approaches are still being studied (at the preclinical and clinical level), being developed or approved. The irremediable but needed changes in the legislation and policies are probably one of the significant causes of delay when actively applying these approaches at a clinical level. In any case, further and more exhaustive clinical studies are necessary to provide heterogeneity and overcome current methodological limitations to confirm the efficacy and safety of most of these proposals. Nevertheless, even though there is a lack of consensus on the potential applicability that each of these strategies may have, the scientific community does agree with the fact that new alternatives are needed in the fight against MDR infections and in the potential application of microbiome modulation to decolonize the gut from MDR bacteria and thus, prevent the development of resistant infections. For that reason, we are confident that in the future, some of these approaches to remove MDR bacteria from gut microbiota, among other sites, will be routinely used.

## Data Availability

Not applicable.

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
