# Peer review of "Insights on Current Strategies to Decolonize the Gut from Multidrug-Resistant Bacteria: Pros and Cons"

_antibiotics, 2023, doi:10.3390/antibiotics12061074_

Round 1

Reviewer 1 Report

Roson-Calero et al presented a review on current strategies to control or eliminate multi-drug resistant bacteria. They gave a brief introduction about the prevalence of MDR and stressed the importance to control MDR, and summarized the available strategies that can potentially eradicate MDR. The review flows pretty well and the perspective is very insightful. Authors should consider the following points as well before publication:

1.    Introduction of the mechanisms of MDR resitant bacteria. Currently it superficially give the terminology of MDR and contamination in communities or hospitals.

2.    Phage or CRISPR method or combined the two methods are very promising and attracting attentions, however, the review does not give too much thoughts. The following citations or relevance can be considered.

Gencay, Y.E., JasinskytÄ—, D., Robert, C. et al. Engineered phage with antibacterial CRISPR–Cas selectively reduce E. coli burden in mice. Nat Biotechnol (2023). https://doi.org/10.1038/s41587-023-01759-y

Wang, R., Shu, X., Zhao, H. et al. Associate toxin-antitoxin with CRISPR-Cas to kill multidrug-resistant pathogens. Nat Commun 14, 2078 (2023). https://doi.org/10.1038/s41467-023-37789-y

DOI: 10.1016/j.cell.2022.07.003

doi: 10.1038/s41586-019-1742-x

Author Response

The point by point answers to reviewers’ comments are stated below. We would like to thank the reviewers for their comments which help in improving the manuscript.

Comments and Suggestions for Authors

Roson-Calero et al presented a review on current strategies to control or eliminate multi-drug resistant bacteria. They gave a brief introduction about the prevalence of MDR and stressed the importance to control MDR, and summarized the available strategies that can potentially eradicate MDR. The review flows pretty well and the perspective is very insightful. Authors should consider the following points as well before publication:

  1. Introduction of the mechanisms of MDR resistant bacteria. Currently it superficially give the terminology of MDR and contamination in communities or hospitals.

We have defined MDR. However to add the mechanisms of resistance would increase a lot the length of the manuscript and this is not the main purpose of this perspective.

  1. Phage or CRISPR method or combined the two methods are very promising and attracting attentions, however, the review does not give too much thoughts. The following citations or relevance can be considered.

Gencay, Y.E., JasinskytÄ—, D., Robert, C. et al. Engineered phage with antibacterial CRISPR–Cas selectively reduce E. coli burden in mice. Nat Biotechnol (2023). https://doi.org/10.1038/s41587-023-01759-y

Wang, R., Shu, X., Zhao, H. et al. Associate toxin-antitoxin with CRISPR-Cas to kill multidrug-resistant pathogens. Nat Commun 14, 2078 (2023). https://doi.org/10.1038/s41467-023-37789-y

Improvement in this section has been carried out and the two abovementioned references has been added.

Reviewer 2 Report

The authors addressed a topic of current interest, given the increasing presence of multi-resistant bacteria. The topic has already been partially addressed by some articles (e.g. Gargiullo 2019, Mascolo 2023), even if the cut given by the authors and the scope of the article is different from the literature published so far. The article is clear and well written; in general I would suggest that authors refer more to systematic reviews and meta-analyses, where available, rather than to individual studies.

Here are reported in detail some suggestions for the authors:

Line 100: please uniform the font size (trimetoprim-sulphametoxazol);

Lines 137-147: please uniform the font size

Chapter: 3.2. Bacteriotherapy: probiotics, prebiotics and synbiotics: a brief scoping review on the topic is also provided by Karbalaei 2022, which includes some RCTs not discussed by the authors.

Chapter 3.3. Fecal microbiota transplantation: a recent meta-analysis is also available on this topic (Tavoukjian 2019)

Lines 309-310: “and the conditions for the optimal efficacy of phages in the gastrointestinal tract are still poorly understood, requiring additional knowledge.”: maybe some information related to the argument could be retrieved by DÄ…browska et al (2019) Systematic Review on the factors shaping phage pharmacokinetics and bioavailability.

Chapter 3.5: CRISPR- Cas Systems: an interesting article reporting in-vivo results for the use of Engineered phage with antibacterial CRISPR–Cas has been just published, and could be of additional interest (Gencay 2023)

Chapter 4: Conclusions: It would be advisable to insert a paragraph reporting among the limitations of the study the use of a non-systematic methodology, which could have led to the loss of pertinent literature.

Relevant literature:

·        Gargiullo L, Del Chierico F, D'Argenio P, Putignani L. Gut Microbiota Modulation for Multidrug-Resistant Organism Decolonization: Present and Future Perspectives. Front Microbiol. 2019 Jul 25;10:1704. doi: 10.3389/fmicb.2019.01704. PMID: 31402904; PMCID: PMC6671974. 

·         Karbalaei M, Keikha M. Probiotics and intestinal decolonization of antibiotic-resistant microorganisms; A reality or fantasy? Ann Med Surg (Lond). 2022 Jul 31;80:104269. doi: 10.1016/j.amsu.2022.104269. PMID: 35958286; PMCID: PMC9358418.

·         Tavoukjian V. Faecal microbiota transplantation for the decolonization of antibiotic-resistant bacteria in the gut: a systematic review and meta-analysis. J Hosp Infect. 2019 Jun;102(2):174-188. doi: 10.1016/j.jhin.2019.03.010. Epub 2019 Mar 26. PMID: 30926290.

·         DÄ…browska K. Phage therapy: What factors shape phage pharmacokinetics and bioavailability? Systematic and critical review. Med Res Rev. 2019 Sep;39(5):2000-2025. doi: 10.1002/med.21572. Epub 2019 Mar 19. PMID: 30887551; PMCID: PMC6767042.

·         Gencay, Y.E., JasinskytÄ—, D., Robert, C. et al. Engineered phage with antibacterial CRISPR–Cas selectively reduce E. coli burden in mice. Nat Biotechnol (2023). https://doi.org/10.1038/s41587-023-01759-y

·         Mascolo A, Carannante N, Mauro GD, Sarno M, Costanzo M, Licciardi F, Bernardo M, Capoluongo N, Perrella A, Capuano A. Decolonization of drug-resistant Enterobacteriaceae carriers: A scoping review of the literature. J Infect Public Health. 2023 Mar;16(3):376-383. doi: 10.1016/j.jiph.2023.01.009. Epub 2023 Jan 16. PMID: 36702012.

·         Ljungquist O, Kampmann C, Resman F, Riesbeck K, Tham J. Probiotics for intestinal decolonization of ESBL-producing Enterobacteriaceae: a randomized, placebo-controlled clinical trial. Clin Microbiol Infect. 2020 Apr;26(4):456-462. doi: 10.1016/j.cmi.2019.08.019. Epub 2019 Sep 5. PMID: 31494254.

Author Response

The point by point answers to reviewers’ comments are stated below. We would like to thank the reviewers for their comments which help in improving the manuscript.

Comments and Suggestions for Authors

The authors addressed a topic of current interest, given the increasing presence of multi-resistant bacteria. The topic has already been partially addressed by some articles (e.g. Gargiullo 2019, Mascolo 2023), even if the cut given by the authors and the scope of the article is different from the literature published so far. The article is clear and well written; in general I would suggest that authors refer more to systematic reviews and meta-analyses, where available, rather than to individual studies.

Here are reported in detail some suggestions for the authors:

Line 100: please uniform the font size (trimetoprim-sulphametoxazol). Done

Lines 137-147: please uniform the font size.  Done

Chapter: 3.2. Bacteriotherapy: probiotics, prebiotics and synbiotics: a brief scoping review on the topic is also provided by Karbalaei 2022, which includes some RCTs not discussed by the authors.

This reference has been added and commented.

Chapter 3.3. Fecal microbiota transplantation: a recent meta-analysis is also available on this topic (Tavoukjian 2019).

This reference has been added and commented.

Lines 309-310: “and the conditions for the optimal efficacy of phages in the gastrointestinal tract are still poorly understood, requiring additional knowledge.”: maybe some information related to the argument could be retrieved by DÄ…browska et al (2019) Systematic Review on the factors shaping phage pharmacokinetics and bioavailability.

This reference has been added and commented.

Chapter 3.5: CRISPR- Cas Systems: an interesting article reporting in-vivo results for the use of Engineered phage with antibacterial CRISPR–Cas has been just published, and could be of additional interest (Gencay 2023)

This reference has been added and commented.

Chapter 4: Conclusions: It would be advisable to insert a paragraph reporting among the limitations of the study the use of a non-systematic methodology, which could have led to the loss of pertinent literature.

 The above-mentioned limitation has been included in the Conclusion Section.

Relevant literature:

  • Gargiullo L, Del Chierico F, D'Argenio P, Putignani L. Gut Microbiota Modulation for Multidrug-Resistant Organism Decolonization: Present and Future Perspectives. Front Microbiol. 2019 Jul 25;10:1704. doi: 10.3389/fmicb.2019.01704. PMID: 31402904; PMCID: PMC6671974. 
  • Karbalaei M, Keikha M. Probiotics and intestinal decolonization of antibiotic-resistant microorganisms; A reality or fantasy? Ann Med Surg (Lond). 2022 Jul 31;80:104269. doi: 10.1016/j.amsu.2022.104269. PMID: 35958286; PMCID: PMC9358418.
  • Tavoukjian V. Faecal microbiota transplantation for the decolonization of antibiotic-resistant bacteria in the gut: a systematic review and meta-analysis. J Hosp Infect. 2019 Jun;102(2):174-188. doi: 10.1016/j.jhin.2019.03.010. Epub 2019 Mar 26. PMID: 30926290.
  • DÄ…browska K. Phage therapy: What factors shape phage pharmacokinetics and bioavailability? Systematic and critical review. Med Res Rev. 2019 Sep;39(5):2000-2025. doi: 10.1002/med.21572. Epub 2019 Mar 19. PMID: 30887551; PMCID: PMC6767042.
  • Gencay, Y.E., JasinskytÄ—, D., Robert, C. et al. Engineered phage with antibacterial CRISPR–Cas selectively reduce E. coli burden in mice. Nat Biotechnol (2023). https://doi.org/10.1038/s41587-023-01759-y
  • Mascolo A, Carannante N, Mauro GD, Sarno M, Costanzo M, Licciardi F, Bernardo M, Capoluongo N, Perrella A, Capuano A. Decolonization of drug-resistant Enterobacteriaceae carriers: A scoping review of the literature. J Infect Public Health. 2023 Mar;16(3):376-383. doi: 10.1016/j.jiph.2023.01.009. Epub 2023 Jan 16. PMID: 36702012.
  • Ljungquist O, Kampmann C, Resman F, Riesbeck K, Tham J. Probiotics for intestinal decolonization of ESBL-producing Enterobacteriaceae: a randomized, placebo-controlled clinical trial. Clin Microbiol Infect. 2020 Apr;26(4):456-462. doi: 10.1016/j.cmi.2019.08.019. Epub 2019 Sep 5. PMID: 31494254.

All references have been included into the manuscript.
